# Implementing allied healthcare professionals in transitional care for older adults with mental health needs: A scoping review

Marina Motsenok[1], Ella C.N. Wong[1], Rozhannaa Sothilingam[1], Monica Antunes[2], Catherine Devion[3], Emma Wilson[3], Amanda Knoepfli[4], Emma Elliot[4], Tracey DasGupta[3], Naomi Ziegler[4], Sander L. Hitzig[1,5,6,7]*

1 St. John's Rehab Research Program - Sunnybrook Research Institute, North York, Ontario, Canada,
2 Faculty of Kinesiology and Physical Education, University of Toronto, Toronto, Ontario, Canada,
3 Sunnybrook Health Sciences Centre, North York, Ontario, Canada, 4 SPRINT Senior Care, Toronto, Ontario, Canada, 5 Rehabilitation Sciences Institute, Temerty Faculty of Medicine, University of Toronto, Toronto, Ontario, Canada, 6 Department of Occupational Science and Occupational Therapy, Temerty Faculty of Medicine, University of Toronto, Toronto, Ontario, Canada, 7 Dalla Lana School of Public Health, University of Toronto, Toronto, Ontario, Canada

* sander.hitzig@sunnybrook.ca

## Abstract

### Introduction

Older adults experience several transitions in care, which can be challenging and stressful. Transitional care ensures continuity of care by addressing patients' healthcare needs before discharge and providing ongoing community support. Although transitional care improves patient outcomes and reduces hospital readmissions, its role in addressing mental health (MH) needs in older adults remains underexplored. This scoping review describes the literature on the integration of allied healthcare (AHC) professionals in transitional care for older adults with MH needs.

### Method

Following PRISMA-ScR guidelines, we analyzed 14 peer-reviewed articles (2010–2024) on transitions for older adults with cognitive impairment, dementia, depression, or suicide risk. Thematic analysis identified key roles, lessons, and patient needs in transitional care provided by AHC professionals.

### Results

AHC professionals, including social workers, occupational therapists, pharmacists, and physical therapists, contributed through discharge planning, physiotherapy, medication reviews, MH counseling, and resource navigation. For caregivers, they provided education on dementia care, behavior management, and psychosocial support, improving caregiver well-being and interactions with persons with dementia.

**Data availability statement:** All data supporting this scoping review are derived from publicly available sources. The search strategy used to identify and select the reviewed manuscripts is fully described in Supporting information, enabling replication of the search and study selection process.

**Funding:** TD, NZ, and SLH were awarded funding from the Slaight Family Foundation. The funder did not play any role in the study design, data collection and analysis, decision to publish, or preparation of the manuscript. SLH holds the John and Sally Eaton Chair in Rehabilitation Research, a joint Hospital-University Chair between the University of Toronto, the Sunnybrook Health Sciences Centre, and the Sunnybrook Health Sciences Centre Foundation.

**Competing interests:** The authors have declared that no competing interests exist.

Transitional care interventions reduced caregiver stress and, in some cases, hospital readmissions. Challenges included suboptimal medication management for persons with dementia. Key facilitators were flexible delivery methods (e.g., telehealth), tailored interventions, and dementia-specific education.

## Conclusion

AHC professionals are vital to transitional care for older adults with MH needs, offering tailored support to patients and caregivers. Enhanced integration, interdisciplinary collaboration, caregiver education, and addressing systemic barriers could improve care quality. Future research should focus on standardizing interventions and optimizing medication management.

## Introduction

Transitional care is a form of healthcare coordination that is provided to patients prior to discharge and continued beyond inpatient care [1], allowing the maintenance of therapeutic relationships that are established during inpatient stays [2]. Healthcare professionals support patients before and beyond the inpatient stay by addressing care needs before discharge and following-up with those needs in the community setting, therefore allowing continuity of care [2]. This includes clinical assessment of patient's healthcare needs and goals, creating care plans, planning and monitoring care and responding to changes in status [3]. Transitional care professionals provide patients with information about available services and guidance, and help patients receive services by linkage, referrals and follow-ups [4]. As such, transitional care plays a critical role in ensuring safe and effective transitions between care settings, particularly for populations with complex and ongoing needs.

Transitional care interventions highlighted in the literature include profession-oriented approaches such as education and training, organizational interventions (including roles like transition nurses, discharge protocols, discharge planning, medication reconciliation, standardized discharge letters, and electronic tools), and interventions focused on patients and their families, such as raising patient awareness, empowerment, and providing discharge support [5]. Various transitional care interventions aimed at ensuring coordinated and continuous healthcare during care transitions have been tested and proven effective in reducing hospital readmissions, shortening hospital stays, and enhancing quality of life and patient satisfaction [6,7]. However, much of this evidence has focused on general medicine populations, with comparatively less attention paid to the mental health (MH) needs of older adults during care transitions.

Transitional care is often provided by allied healthcare (AHC) professionals such as Social Workers (SW) and Occupational Therapists (OT), leading to smoother adjustments, fewer unmet needs [8] upon discharge, and improved functional outcomes [9]. This multidisciplinary support was found to be especially beneficial for persons living with co-occurring MH needs, as it reduces the likelihood of adverse

events that may occur during transitions [10]. Despite the recognized value of AHC professionals in transitional care, limited clarity remains regarding which AHC roles are involved, how they are implemented, and the specific functions they serve for older adults with MH needs during transitions in care.

Older adults with MH conditions such as disturbances in cognition, emotional regulation, or behavior [11], experience physical, emotional, and social strains as a result of their illnesses, as well as by navigating through an often unfamiliar and frequently fragmented healthcare system [12]. Their MH needs may include prevention, early identification, treatment, rehabilitation, and ongoing supports related to MH symptoms, substance use, cognitive or emotional difficulties, and the social determinants that influence mental well-being [13–15]. MH conditions are complex and layered; there is a no one-size-fits all treatment, and they often involve a variety of specialists and care providers [16]. This is a result of the high comorbidity rates of physical health, MH, and substance use issues these patients experience, and an abundance of treatment options. Unfortunately, there is often insufficient cohesion between providers and facilities that comprise the MH care system [17,18]. Corresponding care systems can be geographically distanced, funded by different entities, and care providers may have knowledge of and/or comfort in a specific realm and little interaction with other providers. Additionally, there is a need of sequence in some MH treatments, as certain factors must be addressed in order to target other concerns [17], and a need for efficient communication between professionals focusing on physical healthcare and those focusing on MH [18]. Furthermore, MH symptoms can be overshadowed by chronic illnesses, preventing proper care by MH professionals [19–21].

MH patients could have poorer access to health services – including preventive measures, treatment, and care – compared to the general population. Some patients with MH needs have difficulty taking advantage of available healthcare services and therefore receive poor follow-up for their chronic medical conditions [22]. Indeed, in a qualitative study, interviewed participants with MH concerns emphasized the essential role of accessible healthcare services in maintaining a stable and meaningful life in their homes [4]. Therefore, individuals with MH needs often need help navigating the healthcare system and might lack information about available services [18]. These barriers highlight the importance of targeted transitional care approaches that explicitly address MH-related needs and system navigation challenges for older adults.

. In general, older adults encounter more barriers to successful transitions of care due to higher medical burden and comorbidities, cognitive impairment, and/or frequent polypharmacy [23,24]. Older adults with multiple chronic conditions are more likely to experience depressive symptoms than older adults without these conditions [25], and frequent transitions between hospital and home. These transitions are often poorly coordinated and fragmented, resulting in increased readmission rates, adverse medical events, decreased patient satisfaction and safety, and increased caregiver burden [26]. Furthermore, older adults are disproportionately affected by transitions among and within hospital-based psychiatric and medical settings [23]. Understanding which older adults with MH needs are most vulnerable during transitions and the types of support they would require is essential to improve transitional care outcomes. A pilot quality improvement study indicated that care transitions to an acute medical setting from a psychiatric hospital might be more frequent in the geriatric population than in the other segments of the adult population. Not surprisingly, geriatric patients were overall more medically frail, took more home medications, and experienced longer hospital stays [23]. Therefore, older adults with MH needs are even at more risk for miscommunication in healthcare settings, duplication of services, medical errors and readmission [27].

The current scoping review focuses on transitional care provided to older adults with MH needs. Several knowledge synthesis projects assessed the delivery model and the efficacy of transitional care for older adults with MH needs, such as examining the implementation characteristics in transitional care programs for people living with dementia (PWD) and their caregivers [28], and the appropriateness of such programs for specific healthcare transitions [29]. However, little is known about integrating AHC professionals into transitional care to meet the specific MH needs of older adults. Specifically, there is limited synthesized evidence addressing (1) which older adults with MH needs require transitional care support, (2) the type of role AHC professionals play in supporting these transitions, and (3) they key lessons that can be learned from current implementation efforts.

To address these gaps, this scoping review examines the literature on the implementation of AHC professionals in the field, drawing primarily on studies focused on PWD to provide insights that may also apply to individuals with other MH needs. This review could inform policy and guide interventions to optimize the role of AHC professionals, subsequently improving care, patient well-being and future research on optimizing transitional care for this growing and vulnerable population.

## Methods

### Design

We conducted a scoping review guided by the five step framework proposed by Arksey & O'Malley [30] and Levac, Colquhoun and O'Brien [31], and adhering to PRISMA-ScR (Preferred Reporting Items for Systematic Reviews and Meta-analyses extension for Scoping Reviews Checklist) [32]. The scoping-review process included identifying the research questions, identifying relevant studies, selecting studies, charting the data, and collating, summarizing and reporting the results. We did not undertake the optional step of consulting with key interest groups for this review. A protocol for the review was published in OSF (Open Science Framework; https://osf.io/4aksd).

### Study questions

In this scoping review, we address the following questions:

1. In older adults with MH needs, what are the key lessons in implementing AHC professionals in transition in care?

2. Who are the older adult patients with MH needs requiring supports with transitions in care?

3. What roles do AHC professionals serve in helping older adults with MH needs transition from one care setting to another?

### Search strategy

We consulted with Information Specialists at Sunnybrook Library Services to develop a comprehensive search strategy for literature about transitional care for older adults with MH needs. The search was run without language or date limits on June 27, 2025 in Medline (n = 3,361), Embase (n = 3882), and PsycINFO (n = 1,113) on the Ovid platform and CINAHL (n = 2,570) on the EBSCO platform. The reproducible search strategies are available in Supporting Information. Results files were combined and 2,949 duplicate records were removed using Endnote X9. Covidence software was later used to remove 1,024 additional duplicate records, and 13 additional duplicates were identified and removed manually by the authors (MM, RS, ECNW and MA).

### Search selection

This scoping review includes articles focusing on transitional care programs for older adults with MH needs or the caregivers of older adults with MH needs undergoing care transition. Studies were deemed relevant if they satisfied the following inclusion criteria: (i) published from January 2010 to June 2025, (ii) written in English, (iii) full-text articles available, (iv) and the focus of the article was to address the role of AHC professionals (e.g., OT/PT/SW) in meeting the MH needs of older adults (aged 55 or older) across care transitions. There were no exclusion criteria based on geographic origin; however, review articles were excluded to ensure findings were drawn directly from primary research [33]. Articles describing transitional care programs for older adults with neurodevelopmental or neurological conditions such as autism, stroke, and traumatic brain injury were also excluded as their care pathways usually focus on acute, rehabilitative, or developmental services rather than MH care [34,35]. Three reviewers, ECNW, RS and MA, conducted the study selection, with two reviewers screening each article at every stage (title and abstract screening, as well as full-text). The first author, MM, resolved any discrepancies. The review process was conducted using the Covidence software [36]. Outlines the search and selection process.

## Charting the data and data analysis

The first author (MM) created a standardized data extraction form, which was tested by all authors on a single article. After piloting and refining the form, two of the authors (ECNW, RS or MA) independently reviewed the full-text articles and extracted the data. The first author, who reviewed and consolidated the extracted data and individual reviews, addressed any ambiguities in the extraction process.

The data was charted and summarized to map key study characteristics and findings across the reviewed literature [30]. The first author (MM) and the senior author (SLH) collaborated on the data analysis to inductively identify themes. We applied thematic [38] analysis to evaluate and categorize the included articles, reporting identified themes within the data. The analysis involved a detailed review of the charted data, written observations, discussions of the findings, identification of patterns both across and within studies, categorization of those patterns, and assignment of final theme labels. Validation occurred through iterative review and consensus among reviewers to ensure themes accurately reflected the literature [30].

## Results

The search identified 8,385 peer-reviewed articles. After applying the inclusion and exclusion criteria, 14 peer-reviewed articles were selected for this review (n = 14; see Fig 1). Details of the included articles are presented in Tables 1 and 2. Among the reported studies, five used quantitative designs [39,42,48,50,51], two used a qualitative design [44,43], and seven employed mixed-methods approaches [40,41,45–47,49,52]. Five studies used randomized control trials, while six studies conducted semi structured interviews and focus groups. The studies were conducted in the United States (n = 11), Australia (n = 2), and Sweden (n = 1). Study participants where either caregivers of persons with dementia (PWD) [39,45–47,52], PWD or cognitive impairment [42,44,48,49,50], caregivers and PWD dyads [40,41], older adults [51,49] and veterans [43].

### Older adults with MH needs requiring supports with transitions in care

The reviewed studies mostly focused on older adults living with cognitive impairment, dementia, or specifically, with Alzheimer's disease [39,45–47,52]. However, one study focused on persons experiencing depression [51] and one on veterans at risk of suicide [43]. These older adults transitioned, most commonly, back to the community from a hospital (i.e., acute care, emergency department, rehab), or from a nursing facility. Four articles focused on transitions from community to long term or residential care.

Older adults with MH needs requiring supports with transitions in care had typically complex health needs, often including dementia or other cognitive impairments. These individuals frequently transitioned between healthcare settings, such as hospitals, skilled nursing facilities and home care, where they faced heightened risks of adverse outcomes, including hospital readmissions, emergency department visits, and safety incidents such as falls and medication errors [42,44,40]. They often experienced significant functional limitations, multiple chronic conditions, and dementia-related challenges, such as behavioral symptoms, memory loss, and difficulty following care plans [48,49]. Many relied on family caregivers who often felt unprepared for their caregiving roles, especially during post-discharge periods [45,40]. This population also included veterans and individuals from underserved communities who may lacked access to adequate MH or community-based support services [43,52].

### Roles of AHC professionals in transitional care for older adults with MH needs

AHC professionals delivering transitional care were diverse and included OTs, PTs, SWs, Pharmacists, Paramedics, Psychologists, Family Therapists, Geropsychologists, Case Managers, and Registered MH and Transition counsellors. The reviewed studies emphasize the value of involving diverse healthcare providers in transitional care roles to address the

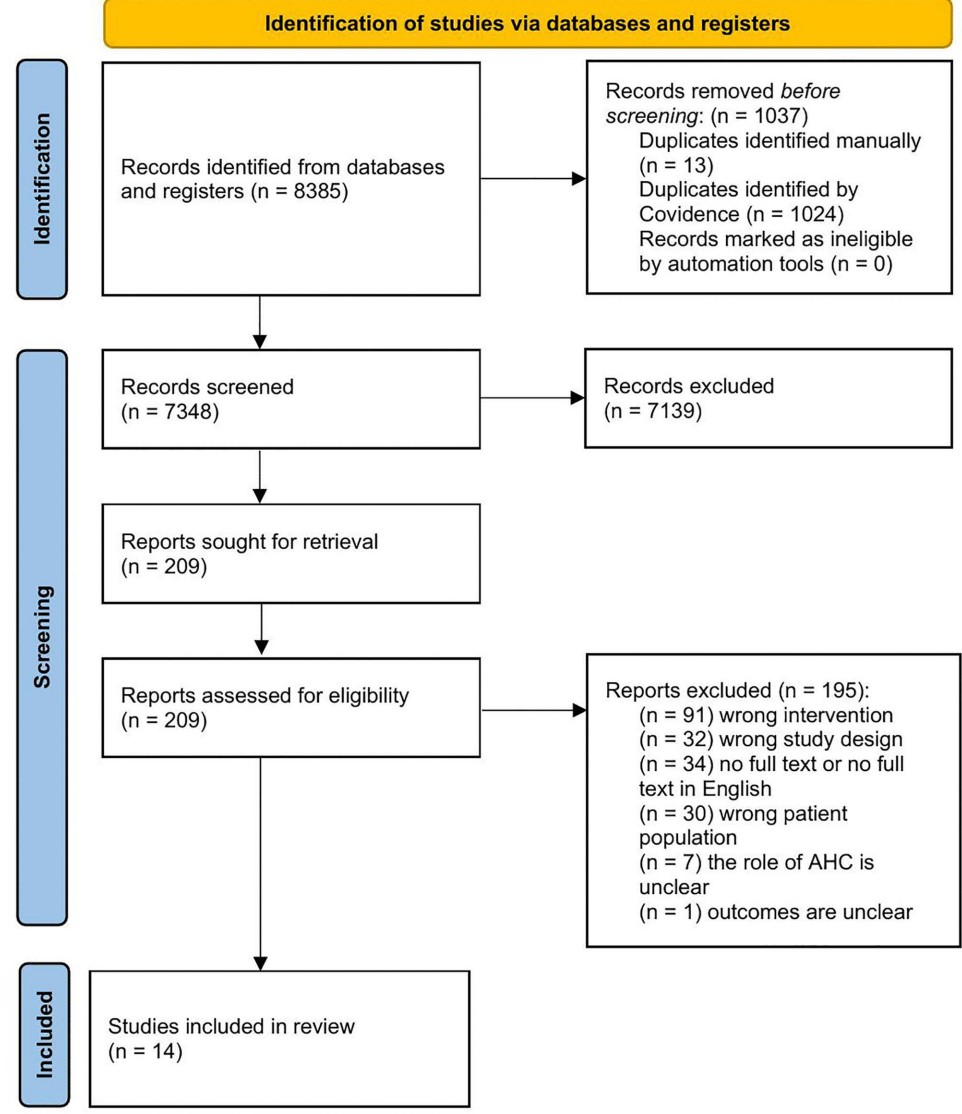

**Fig 1. PRISMA [37].**

complex needs of individuals with MH needs, dementia, and their caregivers. For older adults, AHC professionals delivered discharge planning, physiotherapy, medication review, MH counselling, practical assistance in overcoming barriers in healthcare (i.e., scheduling appointments, communicating with providers), and connected patients with resources. For caregivers and patient-caregiver dyads, AHC professionals provided individualized counselling focusing on education on dementia and skills development (i.e., communication with PWD, promoting safe ambulation, behaviour management, relaxation exercises) and psychosocial support.

SWs and case managers focused on discharge planning, coordinating services, and linking patients to community resources [49]. Palliative care SWs provided counseling and bereavement support [41]. PTs were shown to play pivotal roles in reducing unplanned facility admissions through physiotherapy provided at the patient's home [42]. Pharmacists contributed to medication safety and reduced adverse drug reactions [50,44]. The inclusion of specialized roles, such as

**Table 1. Characteristics of included studies.**

| Authors, reference number, country & design | Objective | Participants & sample size | Outcome measures | Key conclusions |
|---|---|---|---|---|
| Brooks et al., 2024 [39] Australia Randomized control trial (RCT) | To evaluate the feasibility, acceptability, and potential benefits of delivering an individualized counselling program (the Residential Care Transition Module), via videoconferencing to Australian caregivers | Caregivers of PWD $N = 18$ | 1. Perceived Stress Scale (PSS) 2. Caregiver Grief Scale (CGS) 3. Centre for Epidemiologic Studies Depression Scale (CES-D) 4. Caregiver Guilt Questionnaire (CGQ) 5. Geriatric Anxiety Inventory (GAI) 6. Support for Caring subscale of the Adult Carer 7. Quality of Life Questionnaire (SFC) | Intervention has the potential to alleviate feelings of stress and guilt, thereby improving support for caring. Transition counsellors' (TC) skills and dementia-specific knowledge were important mechanisms of impact. Delivery via videoconferencing was acceptable to participants, who appreciated the flexibility in the delivery, and it did not effect establishing therapeutic rapport. |
| Toles et al., 2022 [40] United States (USA) Feasibility study | To conduct a feasibility study of the adapted intervention Connect-Home for Alzheimer's Disease and Related Dementias (Connect-Home ADRD) | PWD and caregiver dyads $N = 19$ | 1. Staff fidelity to the Connect-Home ADRD intervention protocol 2. Acceptability of Connect Home ADRD 3. Preliminary effectiveness outcomes 4. Mechanism through which the intervention addressed the unique transitional care needs of PWD and caregiver dyads 5. Care transitions measure- (CTM-15) 6. Preparedness for caregiving scale (PCS) 7. Re-hospitalizations (three item instrument) 8. Symptoms of dementia | Dementia Caregiving Specialists (OTs) focused on education about fitting care plans with resources in the discharge setting, and the expected impact of Dementia symptoms on the transition. Specialists helped caregivers practice communication skills to engage the PWD with less distress. These skills were used to implement the transition plan (e.g., promoting safe ambulation). Caregivers learned to discern the appropriate level of care when the emerging needs of the PWD exceeded the available resources (e.g., unmanageable safety risks). Specialists supported caregivers with empathy and referrals for additional community and professional care. |
| Toles et al., 2024 [41] USA Mixed methods | To describe the content and quality of transitional care of people with late-stage Alzheimer's diseases and related dementias | PWD and caregiver dyads $N = 40$ | 1. Palliative care plans 2. Patient and caregiver experiences | High-quality, dementia-specific transitional care is achieved when staff have access to resources such as dementia training and care planning templates that enable them to carry the hospital palliative care plan forward into the post-discharge setting, support families in adapting the plan to new circumstances, and help manage the strain and grief associated with changes in health and function. |
| Wang et al., 2020 [42] USA Secondary data analysis | To evaluate the association between home health (HH) services, including skilled nursing (SN), physical therapy (PT), occupational therapy, social work, and homemaking aide assistance with the hazard of unplanned facility admissions among Medicare patients with and without Alzheimer's disease and related dementias (ADRD) | PWD $N = 1,525$; Persons without dementia $N = 4,628$ | 1. Time from HH start of care to an unplanned facility admission of any type 2. AHC contribution to reduce risk of readmission | Skilled nursing and physiotherapy home healthcare reduced hospital readmissions in dementia patients. Compared with the lowest quartile of weekly visits, receiving the highest quartile of SN and PT was related to a reduced hazard of facility admission by 53% (HR ¼ 0.466, 95% CI 0.254, 0.853, P ¼.013) and 86% (HR ¼ 0.140, 95% CI 0.061, 0.320, P < .001) in patients with ADRD, respectively. |

*(Continued)*

**Table 1.** (Continued)

| Authors, reference number, country & design | Objective | Participants & sample size | Outcome measures | Key conclusions |
|---|---|---|---|---|
| Luci et al., 2020 [43] USA Feasibility study | To investigate the effects of a patient navigation intervention (SAVE-CLC) in lowering the risk of suicide in older adult veterans | Veterans $N = 66$ | Feasibility and acceptability of the intervention | A telephone-based intervention addresses an unmet need of integrating MH care into discharge planning through care transitions, and can be scripted for easy standardization across sites and adapted to new settings and levels of expertise. A brief 4-hour training was sufficient for clinicians without prior geriatrics-specific training to successfully administer the intervention. |
| Deeks et al., 2016 [44] Australia Qualitative study | 1. To examine medication management in acute care episodes and care transitions for PWD 2. Create recommendations to improve medication processes | PWD $N = 51$ | 1. Barriers and areas for improvement within medication and care transitions for PWD 2. Medication management | Safe transitions and proper medication management for PWD is complex, often involves addressing behavioral and psychological symptoms commonly associated with the condition, and found to be insufficient, possibly affecting patient safety. Pharmacists should be regularly involved in care transitions, utilizing pharmacist transition coordinators or community liaison pharmacists. The study's findings suggest the need for automatic updates to medication information whenever changes are made in electronic health records, ensuring access for all relevant practitioners. |
| Gaugler et al., 2015 [45] USA Multiple method pilot evaluation | To describe the methods of a pilot study testing the efficacy of the Residential Care Transition Model (RCTM) in helping family caregivers navigate transitioning a PWD into long-term care | Caregivers of PWD $N = 239$ | 1. Frequency, duration, and clinical content of RCTM sessions 2. Dementia Severity 3. Caregiver Stress 4. Depressive Symptoms (Center for Epidemiologic Studies Depression scale [CES-D], Geriatric Depression Scale [GDS] 5. Caregiver Adaptation to Placement | Psychosocial support provided by Transition Coaches (TCs) who helped families manage emotional distress and crises in the months following a PWD's admission to residential long-term care. Greater use of crisis-focused sessions was associated with larger reductions in caregiver depressive symptoms at 4 months (GDS $r = -0.64$; CES-D $r = -0.73$; both $p < 0.01$) and at 8 months (GDS $r = -0.89$, $p < 0.001$). A higher number of sessions addressing physical illness was also associated with reduced caregiver stress at 4 months (role overload $r = -0.60$, $p < 0.05$). Caregivers in the RCTM intervention group reported significantly lower distress at 4 months (NPI-Q, $p < 0.05$) and less caregiver overload at 8 months compared with usual care controls ($p < 0.05$). The TC should have training and experience moderating complex dynamics that may arise during family sessions or crisis events. Transitional care support for caregivers can help them manage key stressors in the months following the transition. |

*(Continued)*

**Table 1.** (Continued)

| Authors, reference number, country & design | Objective | Participants & sample size | Outcome measures | Key conclusions |
|---|---|---|---|---|
| Gaugler et al., 2020 [46] USA Mixed methods | To evaluate the effects of the Residential Care Transition Module (RCTM) on reductions in family caregiver subjective stress and negative MH outcomes | Caregivers of PWD *N* = 239 | 1. Reductions in family member subjective stress and negative MH outcomes 2. Role strains 3. Residential care stress | The required qualifications for the TC role were a Master's degree or higher in marriage and family therapy, social work, counseling, or psychology as well as professional or personal experience working with individuals with dementia and their families. The intervention focused on a key transition, making it targeted and compact. It was not limited by geographic distance, and offered tailored content to meet individual needs, regardless of when the relative was admitted to long-term care. This flexibility enhances its potential for implementation, addressing the diverse needs of families. |
| Gaugler et al., 2024 [47] USA  Mixed methods | To assess the effectiveness of the Residential Care Transition Module (RCTM)—a six-session telehealth intervention that provides psychosocial and psychoeducational support to family caregivers of relatives with cognitive impairment residing in long-term care facilities | Caregivers of PWD *N* = 240 | 1. Caregiver subjective stress and depressive symptoms 2. Secondary role strains 3. Residential care stress 4. Caregiver sense of competence and self-efficacy | The RCTM was delivered by trained TCs who provided structured and ad hoc telehealth sessions emphasizing emotional support, communication strategies, and stress management. Caregivers rated the coaches and sessions highly, describing them as supportive, empathetic, and beneficial for coping and confidence. Although quantitative outcomes showed no significant effects, qualitative feedback suggested that the TCs' relational skills and personalized engagement were central to caregivers' perceived benefits, highlighting the critical influence of provider delivery quality in psychosocial interventions for dementia caregivers. |
| Shah et al., 2022 [48] USA RCT subgroup analysis | To investigate the effect of an adapted Care Transitions Intervention (CTI) in reducing emergency room (ED) readmissions in adults with cognitive impairment | Persons with cognitive impairment *N* = 81 | 1. ED revisits within 14 and 30 days of discharge 2. CTI self-management behaviors (outpatient follow-up, medication self-management, and knowledge of red flags) that target factors associated with effective care transitions | Community-dwelling older adults who had cognitive impairment while in the ED were 75% less likely to revisit the ED within 30 days if received CTI by paramedic coaches (OR 0.25, 95% CI 0.07 to 0.90), who provided informational and practical assistance necessary to overcome barriers like system fragmentation, poor communication with healthcare providers, and limited social support. EDs should implement a process to identify cognitively impaired patients discharged home and provide a tailored care transitions program for those individuals. |
| Prusaczyk et al., 2020 [49] USA Mixed methods | To understand the transitional care actions provided to older adults with and without dementia including what care is being delivered and who is delivering it | PWD *N* = 126; Persons without dementia *N* = 84 | 1. Transitional care actions provided to older adults with and without dementia 2. Types of care providers | Case managers and SWs were the primary providers of discharge planning and care coordination for PWD. SWs may be uniquely suited for these roles because of their ability to handle unexpected social issues that come up for patients during a transition, which may be more common among PWD, who have more complicated transitions. |

*(Continued)*

**Table 1.** (Continued)

| Authors, reference number, country & design | Objective | Participants & sample size | Outcome measures | Key conclusions |
|---|---|---|---|---|
| Gustafsson et al., 2017 [50] Sweden RCT | To examine if the implementation of a comprehensive medication review by a clinical pharmacist within a health care team reduces hospital readmission rates in patients with cognitive impairment or dementia. | PWD or cognitive impairment *N*=429 | 1. Risk of drug related hospital readmissions<br>2. Cost analysis<br>3. Time to institutionalization<br>4. Adherence to quality indicators | The involvement of clinical pharmacists in healthcare teams conducting comprehensive medication reviews did not lead to a significant reduction in the risk of drug-related readmissions among patients with dementia or cognitive impairment. However, after adjusting for heart failure, the intervention showed significant beneficial effects (HR 0.49, 95% CI 0.27–0.90, p=0.02). |
| Aronow et al., 2018 [51] USA Prospective cohort study | To explore differences in delivery of the Coleman Care Transition and subsequent hospital readmissions for older adults who screened positive for depression compared to those who did not | Older adults *N*=4,601 | 1. Re-admissions for those screened positive for depression<br>2. CTI assessment/profile for transitions | SWs played a more directive role for when working with patients with depression compared to those without. When working with this population, SWs encouraged clients to schedule and maintain follow-up appointments, discussed available community resources, and provided referrals to transportation services and home-delivered meals. When addressing MH concerns, SWs were more likely to refer clients to specific or general MH services (12.2% vs. 20.4%, $X^2$=41.85, p<.001, n=4,601), as well as formal case management services (15.4% vs. 61.3%, $X^2$=865.78, p<.001, n=4,601). This approach reflects a pattern of more active guidance and less self-initiated action from older adults who screened positive for depression. This resulted in similar 30-day readmission rates among patients who screened positive for depression and those who did not. |
| Albers et al., 2023 [52] USA RCT | To assess the perceived advantages of the Residential Care Transition Module (RCTM), a psychoeducational/psycho-social, telehealth intervention for caregivers of cognitively impaired relatives living in residential long-term care | Caregivers of PWD *N*=240 | 1. Perceived benefits of the Residential Care Transition Model<br>2. Mood and Emotional Impact<br>3. Changes in perspective<br>4. Confidence in Caregiving | Caregivers benefited from education on dementia progression and behaviour management, personalized resource provision, strategies for communication and engagement with the care recipient, managing multiple roles, and relaxation exercises. Caregivers also benefited from emotional support, expertise, and transition counsellors being a neutral third party during the intervention. These aspects contributed to improved mood, caregiving perspective and confidence, sense of community, and interactions with the PWD and their care team. |

Dementia Caregiving Specialists OTs [40] and Palliative Care SWs [41], showed promise in managing dementia-related risks and enhancing caregiver preparedness through targeted post-discharge support. Multidisciplinary teams facilitated comprehensive care coordination, emotional support, and navigation across care settings, benefiting both patients and caregivers [51,52].

**Table 2. Intervention delivery characteristics.**

| Studies | Transition setting | Clinician delivering the intervention |
|---|---|---|
| Brooks et al., 2024 [39] | Community to residential care | Registered MH counsellors |
| Toles et al., 2022 [40] | Skilled nursing facility to community | OTs, specialized in dementia caregiving |
| Toles et al., 2024 [41] | Acute care to community/long-term care | Palliative Care SWs and RN |
| Wang et al., 2020 [42] | Acute care/rehab to community | PTs, OTs, SWs |
| Luci et al., 2020 [43] | Nursing facility to community | Psychologists (with RN), Geropsychologists |
| Gaugler et al., 2020 [46] | Community to long-term care | Transition counsellors |
| Deeks et al., 2020 [44] | Acute care to community | Pharmacists (hospital and community), OTs |
| Gaugler et al., 2015 [45] | Community to long-term care | Transition counsellors |
| Gaugler et al., 2024 [47] | Community to long-term care | Transition coaches |
| Shah et al., 2022 [48] | ED to community | Paramedic coaches |
| Prusaczyk et al., 2020 [49] | Acute care to community | SW, physician, case manager, RN |
| Gustafsson et al., 2017 [50] | Acute care/rehab to community | Clinical pharmacists |
| Aronow et al., 2018 [51] | Acute care to community | SWs |
| Albers et al., 2023 [52] | Community to long-term care | Transition counsellors |

## Key lessons in implementing AHC professionals in transitional care

Although transition counsellors and SWs were the primary providers of transitional care, there seems to be a lack of consensus regarding the AHC professionals who are best suited to provide transitional care. SWs may have an advantage in transitional care roles due to their ability to manage unexpected social issues that often arise for patients during transitions; challenges that may be more frequent among PWD, who experience complex transitions [49]. Although there was also no consensus regarding the types of transitional care provided by AHC professionals, it was found to improve caregivers' MH by alleviating emotional distress [47,45,40], guilt and stress [39], thereby increasing support for PWD and the quality of shared interactions [39,52,45,49,40].

Mixed results were observed in studies focusing on medication management [50,44], with some subgroup benefits but no consistent reduction in hospital readmissions. Medication management was found to be sub-optimal for PWD during care transitions [44], and did not result, overall, in a significant decrease in the risk of drug-related readmissions for PWD or cognitive impairment [50]. Transitional care that provided practical assistance in overcoming barriers to healthcare reduced emergency department revisits within 30 days for cognitively impaired patients [48], and higher-intensity home health services such as physical therapy, significantly lowered unplanned facility admissions for patients living with dementia [42].

AHC professionals offered customized content and resource support [48,52,46] by referring older adults to additional community and professional services and adapting their assistance to address individual needs. In most cases, AHC professionals utilized phone and video conferencing for transitional care, either exclusively or alongside in-person meetings. This approach appeared acceptable to patients and effective for building therapeutic rapport [39], however, two telephonic interventions lacked clear evidence of clinical outcomes [43,47].

Key identified benefits were flexibility [39,43,46], allowing providers to meet diverse patient and caregiver needs regardless of geographic distance, and adaptability [43,46] by enabling standardized scripts, allowing for easy implementation across sites, adjustments to new settings, and alignment with varying levels of AHC professional expertise. Education was also identified as an important mechanism of impact. This included dementia-specific knowledge [39,48,52,40] (i.e., how dementia symptoms may effect transition, recognising emerging needs), overcoming healthcare related barriers [51,48] such as system fragmentation, and strategies for effective engagement with PWD.

Overall, the reviewed studies highlight that transitional care may be most effective when tailored to individual needs and delivered by a multidisciplinary team. Each AHC professional brings unique expertise, from addressing physical and medical issues to providing psychosocial support and care coordination. Additionally, interventions should align with patient and caregiver specific needs, ensuring continuity and reducing fragmentation across healthcare transitions.

## Studies limitations

The limitations identified across the reviewed studies point to several common challenges in dementia and MH care interventions. Many studies had small sample sizes, which weakened their statistical power and limited the generalizability of their findings [39,52,40]. The use of feasibility and pilot study designs often restricted the ability to rigorously evaluate intervention effectiveness [43,46,40]. Several papers noted short follow-up periods, focusing on immediate or short-term outcomes, thus missing longer-term impacts [39,48,50]. Additionally, many studies relied on homogeneous or narrowly defined samples, often lacking diversity in race, ethnicity, or socio-economic background, which further limited generalizability [51,48,40]. Observational designs in some cases made it difficult to draw causal conclusions and left room for potential confounding factors [42,49]. Other limitations included variability in intervention design [50] care settings and family contexts [47,41], as well as insufficient exploration of underlying mechanisms [48]. The COVID-19 pandemic also disrupted implementation and recruitment in some cases [40], and fragmentation in healthcare systems was a persistent challenge in addressing dementia-specific needs [51,45]. Together, these limitations emphasize the need for larger, more diverse, and methodologically rigorous studies to better understand how to effectively implement AHC professionals in transitional care.

## Discussion

The aim of the present scoping review was to describe the literature on the integration of AHC professionals in transitional care for older adults with MH needs. Overall, this appears to be an under-studied topic, with only 14 identified studies meeting our inclusion criteria. Importantly, the majority of the studies appear to have focused on PWD, with very few focused on other MH concerns in older adults. It should be noted that original intent of this scoping review was not to explore this issue in PWD per se, but studies that included this population were included since PWD frequently experience co-occurring MH issues, such as depression and anxiety [53]. Depression is also a significant risk factor for developing dementia [54]. Some psychiatric conditions are intrinsic to the dementia phenotype, while others, such as major depression, schizophrenia, apathy, and irritability, may precede its onset. Many of these psychiatric disorders can also accelerate dementia progression by increasing distress and disability [55]. Therefore, including dementia within a MH framework provides a more comprehensive understanding of the complex needs of the older adult population, as dementia symptoms often overlap with broader MH challenges.

With regards to points of transition, most of the studies were focused on helping older adults transition back to the community. A noted long-term concern with community transitions for older adults are emotional concerns [56], in that this process is viewed as quite stressful. Follow-up call by psychologists were perceived as beneficial for caregivers' confidence, coping and self-efficacy [47], as well as an effective mechanism for addressing the unmet MH needs of veterans at risk for suicide [43]. There is also evidence supporting both the efficacy and cost-effectiveness of phone follow-ups for supporting transitions in care for patients [57], and has been used to support MH follow-up [58]. Although psychologists may not be cost-effective within some contexts since they are typically more specialized and not funded by the public health system, further work exploring the use of AHC professionals to do follow-ups to meet MH needs during care transitions is worth further investigation.

With regards to transitions to another institutional setting, such as residential care or long-term care, older adults may experience 'relocation stress' [59,60], which is used to describe the MH impact of this type of change. In this instance, older adults may face increased confusion, anxiety, depression, and loneliness during the transition. For those with cognitive impairments, adapting to a new environment can worsen existing challenges, leading to disorientation and social

difficulties [61]. Assimilation into long-term care often takes weeks to months and interdisciplinary teams play a vital role in ensuring smooth and supportive transitions for both residents and families [61].

As caregivers often become de facto coordinators during high-stress care transitions (e.g., liaising with facilities, monitoring quality, and managing family roles), transitional care was found to improve their readiness, coping, and communication capacity [52,46], an often under-recognized coordination pathway in dementia care [52], which may indirectly support more stable placements and fewer crises that trigger acute care use.

Although we only identified 14 studies that met our inclusion criteria, there was a diverse range of AHC providing MH supports to older adults and family members. In some studies, AHC with more extensive training in MH management, such as psychologists, counsellors and SW were used to support transitions. However, other AHC, such as OTs, are also well-suited for supporting MH transitions, and have direct competencies in MH as part of their clinical training [62]. As a result, OTs have been identified as a profession that may be particularly well-suited to support transitions in care [63], and further work exploring their contributions to MH transitions for the older adult population is warranted.

Other AHCs identified included PTs, paramedics, and pharmacists, which are professions that are not primarily focused on MH per se, but have the capacity and ability to provide MH supports as indicated by some of the included studies. For instance, Shah et al., 2022 [48] demonstrated that employing paramedic coaches effectively reduced ED revisits among cognitively impaired older adults. These coaches utilized strategies like motivational interviewing to enhance patients' self-management skills.

The reviewed literature suggests that transitional care could be strengthened by clarifying responsibility for key transition tasks, rather than assuming coordination occurs organically across teams, with SWs and case managers leading discharge coordination and community linkage, and nurses focusing on education and medication safety [49]. Improvements are also likely when medication management is treated as a shared, cross-setting process embedded in team-based care, rather than a standalone discharge activity [50]. As medication management is a common failure point during transitions [50], embedding pharmacists in transition workflows can strengthen cross-setting continuity of medication decisions such as reconciliation, deprescribing, and monitoring plans. In dementia-related transitions, coaching and counselling roles that support caregivers appear to enhance communication, confidence, and informal coordination with providers [52]. Finally, adapting transitional care models to explicitly include cognitively impaired patients may reduce avoidable acute care use and better address the needs of high-risk groups [48].

Several studies have noted that older adults are high users of EDs; [64] many of whom who experience MH challenges [65]. As such, interventions aimed at helping older adults with MH concerns has the potential to lead to more appropriate healthcare utilization, and potential better supports as they transition to the community. As well, there is evidence for training programs in elevating the capacity for MH supports in pharmacists [66], which holds potential as a point of care for MH support in older adults, especially given the high rates of polypharmacy in this population [67]. Regardless, some studies noted a team-based approach, where AHCs worked collaboratively to address MH issues, and likely can address a range of physical, mental and social issues that could negatively impact transitions to other settings.

As noted above, our review identified a number of studies related to meeting the needs of PWD as they either transition back to the community from hospital or to residential or long-term care. A recent systematic review examined navigation programs for PWD and their caregivers [28] indicating diverse providers and offered services, varying in their implementation and targeted outcomes. A consensus about best practices in transitional care is important, since PWD have unique, illness specific challenges such as progressive decline and stigma. Without consensus, programs risk inconsistencies in addressing these unique needs [68].

## Limitations

Our study utilized a broad definition and scope of practice of AHC professionals, which differ between countries, following variations in healthcare systems and workforce structures. While in countries with centralized healthcare systems such

as Australia, AHC professionals are broadly defined, covering a wide array of roles, with typically standardized regulation under a single governing body [69], in the US, operating in a decentralized approach, there is a significant variability in the definitions and regulatory frameworks for allied health professionals across states and institutions [70]. Additionally, there was inconsistency in how transitional care interventions were operationalized across the included studies, leaving some aspects open to the authors' interpretation.

The strength of the evidence is constrained by several important limitations. Many included studies relied on small samples and feasibility designs, limiting statistical power, causal inference, and generalizability [39,48]. Some findings were based on subgroup or post-hoc analyses, which may lead to non-replicable results [50,48]. In addition, outcome measures may not have fully captured the interventions' real-world impact, as qualitative analyses identified meaningful benefits for caregivers that were not reflected in quantitative results [52]. Transferability is further constrained by context-specific factors, as service structures and resources may differ across health systems [43]. Several papers focused on intervention development or study protocols rather than effectiveness outcomes, limiting conclusions about impact [46]. Finally, this review was limited to English-language articles, potentially excluding relevant studies conducted in other languages.

## Conclusions

The reviewed studies highlight the need for scalable, evidence-based interventions targeting both patients with MH needs and their caregivers, with a focus on long-term impacts and diverse populations. This scoping review enhances our understanding of the current state of transitional care and offers insights on optimizing programs for older adults with MH needs.

## Supporting information

**S1 File. Reproducible Search Strategies.**
(DOCX)

**S2 File. PRISMA-ScR Checklist.**
(DOCX)

## Author contributions

**Conceptualization:** Marina Motsenok, Amanda Knoepfli, Emma Elliot, Tracey DasGupta, Naomi Ziegler, Sander L. Hitzig.

**Data curation:** Marina Motsenok, Ella C.N. Wong, Rozhannaa Sothilingam, Monica Antunes, Catherine Devion, Emma Wilson.

**Formal analysis:** Marina Motsenok, Ella C.N. Wong, Rozhannaa Sothilingam, Monica Antunes, Sander L. Hitzig.

**Funding acquisition:** Tracey DasGupta, Naomi Ziegler, Sander L. Hitzig.

**Investigation:** Marina Motsenok, Ella C.N. Wong, Rozhannaa Sothilingam, Monica Antunes, Sander L. Hitzig.

**Methodology:** Marina Motsenok, Catherine Devion, Sander L. Hitzig.

**Supervision:** Sander L. Hitzig.

**Writing – original draft:** Marina Motsenok, Sander L. Hitzig.

**Writing – review & editing:** Ella C.N. Wong, Rozhannaa Sothilingam, Monica Antunes, Catherine Devion, Emma Wilson, Amanda Knoepfli, Emma Elliot, Tracey DasGupta, Naomi Ziegler, Sander L. Hitzig.

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
