## [Decision Letter · Decision Letter 0]

26 Dec 2025

Dear Dr. Motsenok,

Thank you for submitting your manuscript to PLOS ONE. After careful consideration, we feel that it has merit but does not fully meet PLOS ONE’s publication criteria as it currently stands. Therefore, we invite you to submit a revised version of the manuscript that addresses the points raised during the review process.

**ACADEMIC EDITOR:** - please do follow the suggestions of our reviewers and improve the study presentation

We look forward to receiving your revised manuscript.

Kind regards,

Prof. Dr. Dragan Hrncic, MD, PhD

Academic Editor

PLOS One

Journal Requirements:

Reviewers' comments:

Reviewer's Responses to Questions

**Comments to the Author**

1. Is the manuscript technically sound, and do the data support the conclusions?

Reviewer #1: Yes

2. Has the statistical analysis been performed appropriately and rigorously?

Reviewer #1: Yes

3. Have the authors made all data underlying the findings in their manuscript fully available?

Reviewer #1: Yes

4. Is the manuscript presented in an intelligible fashion and written in standard English?

Reviewer #1: Yes

Reviewer #1: Major Issues:

Clarity and Consistency in Terminology: The manuscript occasionally lacks consistency in the terminology used to describe key concepts, such as "transitional care" and "mental health needs." It is recommended to standardize the terminology throughout the manuscript, particularly in the introduction and methods sections, to avoid potential confusion among readers.

Methodology and Data Analysis: While the study adheres to the PRISMA-ScR guidelines, the selection process for articles could benefit from more detail. Specifically, the rationale for excluding review articles and studies focused on conditions outside the scope of dementia (such as stroke or autism) could be better articulated. Furthermore, while thematic analysis is mentioned, a more detailed explanation of how the themes were derived and validated would strengthen the reliability of the results. Clarifying the process of data synthesis could also add transparency to the methodology.

Depth of Discussion: The discussion of the results is relatively superficial in places. While it addresses the roles and challenges faced by allied healthcare professionals (AHC) in transitional care, a deeper exploration of how these roles specifically influence patient outcomes and the potential for improvement in care coordination would be valuable. Furthermore, the limitations of the studies included in the review, such as the small sample sizes and short follow-up periods, should be discussed in greater depth to provide a more balanced view of the evidence.

Minor Issues:

Literature Review and Background: The literature review could benefit from a more structured presentation. While the article references various studies, the connections between the studies and the research questions are not always clear. A clearer link between the review of literature and the gaps identified in the current research would help frame the need for this scoping review.

Inclusion Criteria: The inclusion criteria (published from 2010 to June 2025) are clearly stated; however, a clearer explanation of the exclusion criteria (e.g., the decision to exclude studies on non-dementia neurological conditions) would help improve the reproducibility of the search process.

Statistical Reporting: Although the manuscript includes mixed-methods studies, more detailed statistical reporting of the quantitative studies, especially the effect sizes or statistical significance of key findings, would help enhance the robustness of the conclusions. For example, reporting p-values or confidence intervals for key findings related to hospital readmission rates would be beneficial.

**Do you want your identity to be public for this peer review?** For information about this choice, including consent withdrawal, please see our For information about this choice, including consent withdrawal, please see our Privacy Policy .

Reviewer #1: **Yes:** SALMAN ASHFAQ AHMADSALMAN ASHFAQ AHMAD

---

## [Author Response · Author response to Decision Letter 1]

6 Feb 2026

Reviewer #1: Major Issues:

1. Clarity and Consistency in Terminology: The manuscript occasionally lacks consistency in the terminology used to describe key concepts, such as "transitional care" and "mental health needs." It is recommended to standardize the terminology throughout the manuscript, particularly in the introduction and methods sections, to avoid potential confusion among readers.

We thank the reviewer for flagging this issue. We have standardized the terminology and added clear definitions for key concepts such as mental health conditions and mental health needs (lines 98-103).

2. Methodology and Data Analysis: While the study adheres to the PRISMA-ScR guidelines, the selection process for articles could benefit from more detail. Specifically, the rationale for excluding review articles and studies focused on conditions outside the scope of dementia (such as stroke or autism) could be better articulated. Furthermore, while thematic analysis is mentioned, a more detailed explanation of how the themes were derived and validated would strengthen the reliability of the results. Clarifying the process of data synthesis could also add transparency to the methodology.

We appreciate these valuable suggestions. We added a clarification of our rationale in excluding review articles and stated that this was done to ensure that the scoping review mapped the findings of primary research, following the methodology of scoping reviews as described by Arksey and O’Malley (2005) (lines 195-196). Additionally, we explained why we chose to exclude neurodevelopmental or neurological conditions, as they are typically supported by acute, rehabilitative or developmental services rather than mental health care (lines 198-199). Finally, our data analysis section has been strengthened by a more detailed explanation of how the themes were derived and validated (lines 212-221).

3. Depth of Discussion: The discussion of the results is relatively superficial in places. While it addresses the roles and challenges faced by allied healthcare professionals (AHC) in transitional care, a deeper exploration of how these roles specifically influence patient outcomes and the potential for improvement in care coordination would be valuable. Furthermore, the limitations of the studies included in the review, such as the small sample sizes and short follow-up periods, should be discussed in greater depth to provide a more balanced view of the evidence.

We thank the reviewer for raising these important points. In response, we strengthened the Discussion by providing a more in-depth examination of how allied health care roles influence patient outcomes and enhance care coordination (lines 388–392, 407–419).

Although limitations of the reviewed literature, including small sample sizes and short follow-up periods, were noted in the Results section (lines 315–331), we agree that further consideration was warranted. We have therefore added a dedicated paragraph to the Limitations section to more fully describe the factors that constrain the strength and interpretation of the evidence (lines 447–457).

Minor Issues:

4. Literature Review and Background: The literature review could benefit from a more structured presentation. While the article references various studies, the connections between the studies and the research questions are not always clear. A clearer link between the review of literature and the gaps identified in the current research would help frame the need for this scoping review.

We thank the reviewer for this suggestion. The introduction section has been revised for clarity and to streamline the presentation of the literature and its connection to the research questions.

5. Inclusion Criteria: The inclusion criteria (published from 2010 to June 2025) are clearly stated; however, a clearer explanation of the exclusion criteria (e.g., the decision to exclude studies on non-dementia neurological conditions) would help improve the reproducibility of the search process.

We have clarified our rationale for excluding review articles and non-dementia neurological conditions (lines 195-199).

6. Statistical Reporting: Although the manuscript includes mixed-methods studies, more detailed statistical reporting of the quantitative studies, especially the effect sizes or statistical significance of key findings, would help enhance the robustness of the conclusions. For example, reporting p-values or confidence intervals for key findings related to hospital readmission rates would be beneficial.

We thank the reviewer for this helpful suggestion. Statistical results have been added to Table 1 (Characteristics of included studies) when specific key findings were reported (as opposed to a summary of overall results or conclusions).

---

## [Decision Letter · Decision Letter 1]

16 Mar 2026

Implementing allied healthcare professionals in transitional care for older adults with mental health needs: a scoping review

PONE-D-25-55037R1

Dear Dr. Motsenok,

We’re pleased to inform you that your manuscript has been judged scientifically suitable for publication and will be formally accepted for publication once it meets all outstanding technical requirements.

Kind regards,

Prof. Dr. Dragan Hrncic, MD, PhD

Academic Editor

PLOS One

Additional Editor Comments (optional):

Reviewers' comments:

Reviewer's Responses to Questions

**Comments to the Author**

Reviewer #1: All comments have been addressed

2. Is the manuscript technically sound, and do the data support the conclusions?

Reviewer #1: Yes

3. Has the statistical analysis been performed appropriately and rigorously?

Reviewer #1: Yes

4. Have the authors made all data underlying the findings in their manuscript fully available?

Reviewer #1: Yes

5. Is the manuscript presented in an intelligible fashion and written in standard English?

Reviewer #1: Yes

Reviewer #1: Thank you for revising the manuscript and addressing the reviewer comments. I have carefully examined the revised version along with the authors’ responses to the previous review. The authors have adequately addressed the concerns raised during the earlier round of review, and the revisions have improved the clarity and quality of the manuscript. The responses provided are satisfactory, and the manuscript is acceptable in its current form.

**Do you want your identity to be public for this peer review?** For information about this choice, including consent withdrawal, please see our For information about this choice, including consent withdrawal, please see our Privacy Policy .

Reviewer #1: **Yes:** Salman Ashfaq AhmadSalman Ashfaq Ahmad

---

## [Editor Report · Acceptance letter]

PONE-D-25-55037R1

PLOS One

Dear Dr. Motsenok,

I'm pleased to inform you that your manuscript has been deemed suitable for publication in PLOS One. Congratulations! Your manuscript is now being handed over to our production team.

Kind regards,

on behalf of

Professor Dragan Hrncic

Academic Editor

PLOS One